# The Transcriptional Landscape of Marek’s Disease Virus in Primary Chicken B Cells Reveals Novel Splice Variants and Genes

**DOI:** 10.3390/v11030264

**Published:** 2019-03-16

**Authors:** Luca D. Bertzbach, Florian Pfaff, Viktoria I. Pauker, Ahmed M. Kheimar, Dirk Höper, Sonja Härtle, Axel Karger, Benedikt B. Kaufer

**Affiliations:** 1Institute of Virology, Freie Universität Berlin, Robert von Ostertag-Straße 7-13, 14163 Berlin, Germany; luca.bertzbach@fu-berlin.de (L.D.B.); ahmed.kheimar@fu-berlin.de (A.M.K.); 2Institute of Diagnostic Virology, Friedrich-Loeffler-Institut, Federal Research Institute for Animal Health, Südufer 10, 17493 Greifswald-Insel Riems, Germany; dirk.hoeper@fli.de; 3Institute of Molecular Virology and Cell Biology, Friedrich-Loeffler-Institut, Federal Research Institute for Animal Health, Südufer 10, 17493 Greifswald-Insel Riems, Germany; viktoria.pauker@uni-greifswald.de; 4Department of Poultry Diseases, Faculty of Veterinary Medicine, Sohag University, Sohag 82424, Egypt; 5Department of Veterinary Sciences, Institute for Animal Physiology, Ludwig-Maximilians-Universität München, 80539 Munich, Germany; sonja.haertle@tiph.vetmed.uni-muenchen.de

**Keywords:** Marek’s disease virus (MDV), RNA-seq, transcriptome, splicing, polycistronic viral transcripts, primary B cells, RB1B, CVI988/Rispens, ICP0

## Abstract

Marek’s disease virus (MDV) is an oncogenic alphaherpesvirus that infects chickens and poses a serious threat to poultry health. In infected animals, MDV efficiently replicates in B cells in various lymphoid organs. Despite many years of research, the viral transcriptome in primary target cells of MDV remained unknown. In this study, we uncovered the transcriptional landscape of the very virulent RB1B strain and the attenuated CVI988/Rispens vaccine strain in primary chicken B cells using high-throughput RNA-sequencing. Our data confirmed the expression of known genes, but also identified a novel spliced MDV gene in the unique short region of the genome. Furthermore, *de novo* transcriptome assembly revealed extensive splicing of viral genes resulting in coding and non-coding RNA transcripts. A novel splicing isoform of MDV UL15 could also be confirmed by mass spectrometry and RT-PCR. In addition, we could demonstrate that the associated transcriptional motifs are highly conserved and closely resembled those of the host transcriptional machinery. Taken together, our data allow a comprehensive re-annotation of the MDV genome with novel genes and splice variants that could be targeted in further research on MDV replication and tumorigenesis.

## 1. Introduction

Marek’s disease virus (MDV), also known as Gallid alphaherpesvirus 2, causes a deadly lymphoproliferative disease in chickens. Typical clinical symptoms include immunosuppression, paralysis and polyneuritis, acute brain edema, and lymphoma that develop as early as 3 weeks post infection [1,2]. MDV has a big economic impact on the poultry industry worldwide due to animal losses, reduced growth, decreased egg production, and cost of vaccination [3]. Vaccines are crucial for the protection against MDV, as very virulent strains can cause mortalities of up to 100% in susceptible unvaccinated chickens [4]. Live attenuated MDV vaccines such as the gold standard Rispens strain (CVI988) are highly effective in preventing tumor formation [3,5], but do not provide sterilizing immunity.

MDV infection is initiated by the inhalation of the virus from a contaminated environment. Macrophages and other phagocytic cells are thought to take up the virus and transfer it to lymphatic tissues, where B and T cells are infected [6]. B cells are efficiently infected during the initial lytic virus replication, whereas T cell subsets play key roles in MDV pathogenesis [7,8,9,10,11]. MDV establishes latency predominantly in CD4+ T cells, which can also transport the virus to the feather follicle epithelium. These cells efficiently produce the infectious virus and shed it into the environment [3]. Latently infected T cells can also be transformed, resulting in deadly lymphomas. 

MDV has a 180 kilo base pairs class E genome consisting of a unique long (U_L_) and a unique short (U_S_) sequence that are flanked by terminal (TR_L_ and TR_S_) and internal (IR_L_ and IR_S_) inverted repeat regions [12,13]. MDV encodes about 100 proteins that orchestrate the virus life cycle and/or contribute to pathogenesis [12,14]. Until now, analyses of the viral transcriptome has been limited to chicken fibroblasts that are not infected in chickens, ex vivo samples [15,16] and tumor cells [17]; however, the mRNA expression in the primary target cells of lytic replication in vivo remained elusive. This is mainly due to the short lifespan of B and T cells in culture and the low quantity of infected cells in lymphoid organs of chickens [7,18]. To overcome this obstacle, we recently developed an in vitro infection system for primary B and T cells that allows for a prolonged survival and efficient infection of these cells [19]. We used this system to analyze the MDV transcriptome in the most frequent lytically infected cell type in vivo, the B cells.

In this study, we performed next generation RNA-sequencing and protein profiling in primary B cells infected with the very virulent RB1B strain or the vaccine strain CVI988. Our data reveal that the coding capacity of the MDV genome is larger than expected. We identified novel MDV genes and splice variants, and confirmed them either on the protein level or by RT-PCR. This comprehensive approach provides novel insights into the transcriptome of MDV in the natural target cells and supply a basis for future research on MDV pathogenesis.

## 2. Materials and Methods 

### 2.1. Ethics Statement 

Valo specific-pathogen-free (SPF) chickens (VALO BioMedia GmbH, Osterholz-Scharmbeck, Germany) were housed for 6 to 11 weeks of age and humanely euthanized prior to the isolation of the bursa of Fabricius. The animal work was approved by the governmental agency, the Landesamt für Gesundheit und Soziales (LAGeSo) in Berlin, Germany (approval number T0245/14, approval date 23 October 2014). 

### 2.2. Cells 

Embryonated SPF Valo chicken eggs (VALO BioMedia GmbH,) were used for the preparation of chicken embryo cells (CEC). CEC were maintained in minimal essential medium (MEM, PAN Biotech; Aidenbach, Germany) supplemented with 1-10% fetal bovine serum (FBS) and penicillin/streptomycin as previously described [20]. B cells were obtained from the bursa of Fabricius by dissociation of the organ and subsequent isolation of the cells by density gradient centrifugation as previously described [21]. Briefly, the bursa of Fabricius was homogenized through a 40 μm cell filter to obtain a uniform single cell suspension. Suspension cells were carefully applied on a Biocoll separating solution (Biochrom; Berlin, Germany), centrifuged for 12 min at 650 × g with slow acceleration, and deactivated deceleration. Lymphocytes at the interphase were carefully transferred to a new tube, washed with PBS, and maintained in RPMI 1640 (PAN Biotech) supplemented with 10% FBS and penicillin [100 U/mL]/streptomycin [100 µg/mL] at 41 °C under a 5% CO_2_ atmosphere. B cells were activated using recombinant soluble chicken CD40 ligand (chCD40L) [22], which was expressed in HEK293 cells and purified using a Vivacell 250 ultrafiltration concentrator (Sartorius; Göttingen, Germany).

### 2.3. Viruses 

All viruses were reconstituted by calcium phosphate transfection of CEC with purified bacterial artificial chromosome (BAC) DNA as previously described [23]. The very virulent RB1B strain and the vaccine strain CVI988 both express a green fluorescent protein (GFP) under the control of the early thymidine kinase promotor. The viruses were propagated on CEC for up to six passages, and infected cells were stored in liquid nitrogen until further use. All virus stocks were titrated on fresh CEC.

### 2.4. Infection of Primary Chicken B Cells

Primary chicken B cells were infected by co-cultivation with infected CEC due to the strictly cell-associated nature of MDV. One million CEC were infected with 30,000 plaque-forming units (PFU) of CVI988, RB1B, or mock infected. After 4 days, one million B cells were seeded on the infected 6-well-plates in the presence of CD40L for 16 h at 41 °C. All B cells were then carefully removed from the CEC monolayer, washed with phosphate buffered saline (PBS), and prepared for fluorescence-activated cell sorting.

### 2.5. Flow Cytometry

Viable bursal B cells were detected using the eFluor780 fixable viability dye at a dilution of 1:1000 (Affymetrix eBioscience; San Diego, CA, USA) as previously described [19]. Cells were sorted using a FACS Aria III sorter and the FACSDiva software (Becton Dickinson; Franklin Lakes, NJ, USA). For each sample, approximately 10^5^ to 10^6^ infected B cells were sorted at 4 °C and stored at −80 °C until further analysis. The purity of GFP+ sorted fractions was determined by FACS reanalysis and yielded 99.73% (±0.46 SD) for mock-infected cells, 95.33% (±1.29 SD) for RB1B infected cells and 97.47% (±1.33 SD) for CVI988 infected cells.

### 2.6. High-Throughput RNA-Sequencing

RNA was isolated from three independent experiments of CVI988 or RB1B infected chicken B cell cultures and sequenced as described [24]. Briefly, total RNA was extracted using TRIzol reagent (Life Technologies; Carlsbad, CA, USA) in combination with the RNeasy Mini Kit (Qiagen; Hilden, Germany) following the manufacturer’s instructions. Additionally, RNA was treated with DNase using the RNase-Free DNase Set (Qiagen). Subsequently, ERCC ExFold RNA Spike-In mix 1 (Invitrogen; Carlsbad, CA, USA) was added to the total RNA as an internal control and the polyadenylated (poly(A)) RNA fraction was extracted using the Dynabeads mRNA DIRECT Micro kit (Invitrogen). Whole transcriptome libraries were prepared using the Ion Total RNA-Seq Kit v2 (Life Technologies) following the manufacturer’s instructions. Quality and quantity of the nucleic acids was controlled at each step using the NanoDrop 1000 spectrophotometer (Peqlab) or Agilent 2100 Bioanalyzer (Agilent Technologies; Böblingen, Germany) in combination with appropriate chips, respectively. The resulting libraries were finally quantified using the KAPA Library Quantification Kit for Ion Torrent (Kapa Biosystems; Wilmington, MA, USA) on a CFX96 Real-Time PCR Detection System (BioRad Laboratories) and sequenced on an Ion S5XL system (Life Technologies) using the Ion 540 OT2 and Chip kit (Life Technologies).

### 2.7. Sequence and Bioinformatic Analyses 

Reads from separate sequencing runs (technical replicates) of the same library (biological replicates) were combined and quality-trimmed using the 454 Sequencing System Software (v. 3.0; Roche; Mannheim, Germany) along with appropriate Ion Torrent specific adapter sequences. Each quality-trimmed data set was then mapped to a non-redundant version of the MDV reference NC_002229.3 [12] (only segments U_L_-IR_L_-IR_S_-U_S_) using STAR (version 2.6.1a; [25]), running in basic two-pass mode. In this manuscript, the MDV genes were designated according to the current gene nomenclature used for the prototype alphaherpesvirus herpes simplex virus 1 (HSV-1) [26,27]. As the mRNA libraries were amplified for several rounds, the sequence duplicates were removed prior to *de-novo* assembly and coverage analysis. Therefore, the duplicated reads were marked and removed in each mapped dataset using Picard (version 2.18.20; http://broadinstitute.github.io/picard). Subsequently, the unique aligned reads were directionally sorted using samtools (version 1.9; [28]) and the sequence depth was deduced from each dataset using bedtools (version 2.15.0; [29]) and samtools. The directionally sorted reads were then used for *de-novo* assembly using the 454 Sequencing System Software (v. 3.0; Roche) running in “-cdna” mode. Assembly was done for each biological replicate separately and with the combined read data from all replicates. Deduplicated and directionally sorted reads were also used as basis for the coverage plots. In order to receive high quality splice junctions, all assembled “isotigs” (transcript variants) were then re-mapped to the non-redundant version of MDV reference NC_002229.3 [12] using STARlong (version 2.6.1a; [25]) in basic two-pass mode. Options were set to allow a single mapping isotig to yield a splice junction with a maximum intron length of 10,000 bp. Only positions corresponding to the high-quality splice junctions were then selected from the splice junctions of the initial mapping approach for further analysis. The frequency of spliced reads was calculated by dividing the number of reads with splice junction by the total number of reads at the respective donor site. The overall splice frequency at a single donor site for CVI988 and RB1B was then averaged from the individual splice frequencies in the replicates. Based on the deduced splice junctions, the positions up- and downstream of these were extracted and visualized using the R (version 3.4.1; [30]) package “ggseqlogo” (version 0.1; [31]) in combination with RStudio (version 1.0.153) in order to receive information on donor and acceptor motif sequences. Polyadenylation cleavage clusters were determined with ContextMap (version 2.7.9; [32]) in combination with bowtie2 (version 2.2.9; [33]) using the complete trimmed dataset as input and the “--polyA” parameter. The resulting poly(A) cleavage sites were then combined into clusters, as the exact position of mRNA cleavage downstream of a cleavage signal can be heterogeneous [34]. Starting with the first poly(A) cleavage site, all other sites on the same strand within a window of the next 30 nt were combined into a single cluster. The window was then moved to the next cleavage site that was not within the last cluster. This was repeated for all cleavage sites. To scan for enriched regulatory motifs within the three prime untranslated region (3′-UTR) of MDV transcripts, the sequences 50 nt up- and 20 nt downstream of the start position of each identified poly(A) cleavage cluster were extracted and analyzed using DREME as part of the MEME suite (version 4.9.0; [35]) using default settings. Subsequently, the identified enriched motif (AWTAAA) was searched in the non-redundant version of MDV reference NC_002229.3 [12] using FIMO (version 4.9.0; [36]) with default settings. Poly(A) cleavage clusters and regulatory motifs were then grouped into relevant pairs based on the FIMO *p*-value and the their distance, allowing a maximum distance of 50 nt. Differential gene expression between RB1B and the vaccine strain CVI988 was conducted using Salmon (version 0.12.0; [37]) in combination with DESeq2 (version 1.18.1; [38]) as described earlier [24]. All relevant MDV CDS sequences were used as transcript reference and genes with an adjusted *p*-value > 0.01 were considered significant. Potential phosphorylation sites in novel protein SORF6 were predicted using the NetPhos 3.1 Server [39] and its DNA-protein binding probability using the DNABIND server [40].

### 2.8. LC-MALDI TOF/TOF Mass Spectrometry 

Infected FACS sorted primary chicken B cells and mock infected primary chicken B cells were lysed in batches of 1.5 × 10^6^ cells using 150 µl of a lysis buffer containing 0.1 M DL-Dithothreitol (DTT) and 2% SDS in 0.1 M Tris-HCl (pH 8.0) at 99 °C for 5 min. Protein contents were determined by densitometry of Coomassie stained SDS gels [41,42]. After cell lysis, 20 µg aliquots were digested using the FASP protocol as described [43]. Samples were differentially labeled by dimethylation [44] using unlabeled and ^13^C-labeled formaldehyde, respectively, and subjected to nano-LC MALDI-TOF/TOF mass spectrometry as described previously [45]. Briefly, peptides were separated by nano reversed-phase liquid chromatography (EASY-nLC II, Bruker; Bremen, Germany), spotted to a MALDI target (Proteineer fcII, Bruker), and analyzed with an UltrafleXtreme MALDI-TOF/TOF mass spectrometer (Bruker) as described previously [46]. Peptide spectra were acquired in the m/z range 700 to 3.500 Da with a minimum signal-to-noise (S/N) ratio of 7. Proteins were identified with a Mascot server (version 2.4.1; Matrix Science Ltd; London, UK) and analyzed using ProteinScape software (version 3; Bruker). Oxidation of methionine, acetylation of protein N-termini, and dimethylation of lysine and peptide N-termini with either isotopomer were set as variable modifications, whereas the carbamydomethylation of cysteine residues was set as a fixed modification. Two independent experiments were performed with inverted labeling. As database for the protein identification with the MASCOT search engine (Matrix Science Ltd), the *Gallus gallus* proteome was downloaded from the ENSEMBL website [47] and the viral sequences were added to the FASTA file. Viral protein content was calculated in mol% using the exponentially modified protein abundance index (emPAI) [48]. To identify peptides covering the potential new splicing sites discovered by RNA-sequencing, a database with sequence fragments covering a 67 amino acid region centered on the splicing site was constructed and used for the database search with MASCOT.

### 2.9. Reverse Transcription and PCR over Splice Junctions

RNA was isolated as described above. cDNA was synthesized after DNAse treatment (Promega; Fitchburg, WI, USA) with the Applied Biosystems High-Capacity cDNA Reverse Transcription Kit (Thermo Fischer; Waltham, MA, USA). Conventional Taq-PCR was performed with primers specific to the respective viral gene (Appendix A). Amplification of BAC DNA was used as a positive control. Mock-infected cells and samples without reverse transcriptase to exclude a contamination with genomic DNA were included as negative controls.

### 2.10. Data Availability

The RNA-seq raw data were deposited in the ArrayExpress database at EMBL-EBI (www.ebi.ac.uk/arrayexpress) under accession number E-MTAB-7772. A supplementary GFF file for the reference sequence NC_002229.3 containing annotations for all detected introns, poly(A) cleavage sites and associated motifs, as well as the novel CDS for SORF6 can be found in the Appendix A.

## 3. Results and Discussion

### 3.1. The MDV Transcriptional Landscape

To assess the transcriptional landscape of MDV in the primary target cells of lytic replication in vivo, we used a previously established in vitro infection system for primary chicken B cells [19]. B cells were infected for 16 h with the very virulent MDV strain RB1B and the vaccine strain CVI988 and analyzed by high-throughput RNA sequencing. 

The overall RNA-seq dataset consisted of 82.6 million reads from three biological replicates of CVI988 (48.2 million reads) and two biological replicates of RB1B (34.4 million reads). A third replicate of RB1B did not yield sufficient amount of reads and was therefore excluded from our analysis. An average of 10.5% and 11.2% of the CVI and RB1B datasets respectively could be mapped to the MDV reference sequence (see Appendix A). 

The position and direction of mapped reads fitted very well to the previously annotated MDV genes (Figure 1A). Highly abundant genes like the immediate early gene SORF1 (ICP4) or the UL49 tegument protein (VP22) correlate well with previously published data [49]. Surprisingly, only minor differences were detected between the transcriptome of CVI988 and RB1B. Comparing the 94 detected MDV genes, two variants of MDV075 encoding the 14-kDa polypeptides (pp14), were significantly higher expressed in RB1B infected primary B cells (Figure 1B). These phosphorylated cytoplasmic proteins arise from splice variants of the same gene and are thought to be involved in transcriptional regulation and increased neurovirulence [50,51,52]. Furthermore, the hypothetical gene MDV082 [53] that is located on the same transcript as the ICP4 gene, was significantly higher expressed in CVI988 infected B cells. However, the transcriptome of RB1B and CVI988 only shows subtle differences in primary chicken B cells, suggesting that the differences in their pathogenesis might be due to sequence changes on the protein level and/or functional differences of virulence factors. Similarly, we only detected marginal quantitative differences between the expression levels of viral proteins in a proteome analysis using LC-MALDI TOF/TOF mass spectrometry (Appendix A and Appendix A). Taken together, our data indicate that there are only minor differences in the mRNA and protein expression levels after B cell infections with the very virulent RB1B or the CVI988 vaccine strain. 

### 3.2. Splicing of Polycistronic MDV Transcripts

In addition to the transcriptional profile, we could readily identify 71 introns that were represented by at least one *de novo* transcript (Figure 2A). Some of the introns and associated spliced genes have been previously described such as the viral lipase (vLIP) [54], LORF2 (MDV012) [55], UL15 [12], UL44 (glycoprotein C) [56], vIL8 [57] and pp14 [51]. However, analysis of the MDV transcriptome revealed a number of novel splice forms (Appendix A). These results are in line with previous RNA-seq analysis for other alphaherpesviruses [58] that also revealed a plethora of novel splice products. The detected splice variants could contribute to viral proteomic diversity and could prevent viral mRNA degradation through the virion host shutoff UL41 endoribonuclease [59]. For HSV-1 it has been shown that UL41 not only targets many cellular but also viral mRNAs. Spliced mRNAs are protected from UL41-mediated degradation by bound exon junction complexes (EJCs) [60].

The identified splice site sequences mostly represent canonical splicing motifs, containing the GT at the donor and AG at the acceptor sites (Figure 2B,C). The intron length varied between 70 and 8651 nt (Figure 2E). Intriguingly, the intron frequencies differed between RB1B and CVI988 in infected primary chicken B cells (Figure 3, Figure 4 and Figure 5). By matching the intron positions with our MS data, we identified a peptide that spans the exon-exon junction of UL15 (Appendix A).

The analysis of poly(A) cleavage signals within the RNA-seq data revealed abundant bicistronic and polycistronic MDV transcripts (Appendix A). These transcripts encode for two or more proteins and were characterized as regions of high coverage that were not separated by a poly(A) cleavage site. Here we found that the canonical AATAAA motif is the most frequent and functional polyadenylation signal in MDV, followed by ATTAAA (Figure 2D). Interestingly, we also found evidence for alternative non-canonical polyadenylation signals in MDV mRNA 3′ UTRs (Figure 2 and Appendix A). The distances between the detected AWTAAA polyadenylation signal motif and the actual poly(A) cleavage site was 13.8 nt (±4.4 SD) and confirmed that not only the polyadenylation signal sequence, but also its distance from the poly(A) is highly conserved (Figure 2D,F) [61,62].

### 3.3. The Transcriptional Makeup of the MDV Unique Long Region (U_L_).

The MDV unique regions mainly harbor genes that are conserved among alphaherpesviruses and are involved in DNA replication and production of progeny virus [26]. The U_L_ spans over roughly 113,000 base pairs and harbors the majority of the MDV-encoded genes [26]. Within the U_L_, we could detect high transcription rates of nearly all annotated genes. Splicing was identified in multiple genes including RLORF14 (pp24), vLIP [63], and LORF2 (MDV012) [55] and in a transcript antisense to UL5 (MDV017) (Figure 3). Only minor differences were observed in the intron frequencies between RB1B and CVI988. To confirm the splice events and frequencies detected by RNA-seq, we performed RT-PCR analyses on several randomly selected genes (Figure 3C, Figure 4B, Figure 5C, Appendix A). All analyzed genes showed a comparable splice pattern in both RNA-seq and RT-PCR. In addition, we confirmed a novel splice site of UL15 by MS and RT-PCR (Appendix A). UL15 encodes the tripartite terminase subunit that is involved in DNA packaging into the viral capsid. Splicing of UL15 mRNA has already been observed in herpes simplex virus type 1 (HSV-1) [64] and duck enteritis virus (DEV) [65]. However, the observed UL15 isoforms in MDV are to our knowledge unknown and expand the number of potential proteins encoded by UL15 to at least five.

We could also confirm known splice sites in UL44 (gC) in our analysis (Figure 4) [56]. These splice variants lead to a gC protein that lacks the transmembrane domain and is secreted into the supernatant [56]. Beyond that, we also confirmed novel splice sites like in RLORF14a (pp38) (Appendix A). Several capsid and tegument protein-encoding genes encoded in the U_L_ and U_S_ regions, like UL18 (triplex capsid protein 2), UL19 (VP5), UL49 (VP22), UL49.5 (gN), US7 (gI) and US8 (gE) also undergo different splicing events (Appendix A), contradicting the long-standing paradigm that splicing is a rare phenomenon throughout the alphaherpesvirus family [13]. 

Some identified splice variants would result in proteins with an altered membrane topology (TMHMM Server, v. 2.0). This is for example the case for splice variants of pp24 and of pp38, which results in changes of the previously assessed hydrophobic anchor domains of both proteins [66]. While pp38 seems to exist as splice variants with and without a membrane anchor, splicing of pp28 could retain its membrane association while altering its function (Appendix A).

### 3.4. The Transcriptional Makeup of the MDV Unique Short Region (U_S_).

The MDV U_S_ region contains many genes that play important roles in the viral life cycle. Intriguingly, we detected splice variants of several envelope glycoproteins as described above. In addition, we identified a hitherto uncharacterized spliced transcript of a gene located downstream of SORF2A, termed SORF6 (Figure 5). This novel gene possesses an upstream TATA box in the transcriptional regulatory region, an intron and exon with respective donor and acceptor site and a downstream polyadenylation signal with the poly(A) cleavage cluster (as described in Figure 2). Furthermore, the resulting protein is predicted to be 85 amino acids in size, shows several predicted phosphorylation sites (Appendix A) and may act as a DNA-binding protein (DNABIND server [40]). The region containing this novel gene (Figure 5) has previously been associated with the virulence of the virus [67]. However, more work needs to be done to understand the contribution of this novel gene and the region in MDV virulence.

### 3.5. The Transcriptional Makeup of the MDV Repeat Regions

The repeat regions mostly contain MDV-specific genes encoding for proteins or RNA that play a role in the cell tropism, MDV pathogenesis, latency, and transformation [14]. Here, we observed excessive splicing antisense to ICP4. These transcripts are part of the latency associated transcript (LAT) region, have a complex splice pattern, and their functions remain largely unknown [68,69]. Some of these RNAs function as MDV-encoded micro RNAs and are described elsewhere [70,71].

Only moderate RLORF7 (Meq) and vIL8 splicing was detected in infected primary B cells 16 hpi although more extensive splicing activity has been observed in this region of the MDV genome in vitro and in vivo [72,73,74]. These splice variants are likely higher expressed in latently infected and transformed cells. The splice variants of the neurovirulence factor pp14 encoded by MDV075 were efficiently detected as published previously [15,44]. 

In the RNA-seq data, we did not detect any reads complementary to vTR. However, this region is annotated as a hypothetical MDV gene termed RLORF1 (an arginine-rich protein/ICP0-like protein) [75]. RLORF1 is discussed as a potential positional orthologue of alphaherpesviral ICP0 proteins; however, it does not contain typical ICP0 features such as a C3HC4 zinc RING finger at the N-terminus or a nuclear localization signal (NLS). To assess if ICP0 protein is expressed and if it plays a role in replication, we generated recombinant MDV mutants harboring an HA-tagged ICP0 (RB1B_ICP0-HA) or an ICP0 knockout (RB1B_ΔMetICP0). The knockout did not affect MDV replication and cell to cell spread in vitro (Appendix A) and no ICP0 was detected by western blotting (Appendix A), suggesting that ICP0 is not expressed and therefore does not play a role in the virus life cycle.

Furthermore, we detected several poly(A) cleavage sites in combination with appropriate motifs within the MDV repeat regions, that are to our knowledge undescribed. The presence of these transcriptional signals in combination with sufficient read coverage suggest the existence of hitherto hypothetical protein coding regions, including RLORF11 and MDV082.

### 3.6. MDV Noncoding RNAs

Although we enriched for poly(A) mRNA, some newly identified introns do not lie in annotated MDV coding sequences and some splice donor and acceptor sites do not give rise to conclusive protein-encoding mRNAs. Such sequences could be easily regarded as ‘nonsense’ transcripts that are rapidly degraded. However, the importance of noncoding RNAs (ncRNAs) in MDV infections is expanding. Several viral ncRNAs were found to be expressed by MDV [75,76,77,78] but the multitude of functions played by viral ncRNAs, and especially by long ncRNAs (lncRNAs) and stable intronic sequence RNAs (sisRNAs), have not been thoroughly investigated yet and the unexpected transcriptomic complexity may have been overlooked in MDV research so far. Of note is, that antisense transcription was also observed in related alphaherpesviruses like HSV-1 or pseudorabies virus (PRV), and in human herpesvirus 6 (HHV-6) RNA-seq data [79,80,81].

Taken together, our MDV RNA-seq data provide novel insights into the transcriptional profile of the RB1B and CVI988 strains. Despite stark differences in their pathogenicity, the two viruses show a similar transcriptomic profile in primary chicken B cells.

## 4. Conclusions

B cells are a major target for lytic MDV replication in vivo [8,10]; however, it remained impossible to assess the MDV transcriptome in primary B cells, due to the short-lived nature of these cell. The aim of this study was to evaluate the gene expression profiles of the very virulent RB1B strain and the commercial MDV vaccine CVI988 in primary chicken B cells by RNA-seq using our recently established in vitro infection system [19]. We developed a bioinformatics pipeline that can be easily transferred to other herpesviruses or large DNA viruses to identify unknown transcript isoforms and associated motifs.

The RNA-seq revealed the expression of 94 MDV transcripts and the presence of 71 introns that lead to mostly novel splice forms and antisense transcripts. In addition, we could identify a novel gene in the U_S_ region of the MDV genome that we will characterize in future studies. While some of the detected splice sites were previously published, we identified several novel splice variants and confirmed some of them by RT-PCR and/or MS. However, more work is certainly required to dissect their relevance in the MDV life cycle. 

We found that MDV produces bicistronic and polycistronic transcripts as a mechanism to maximize its coding capacities. Poly(A) cleavage after the upstream AATAAA motif seems to be the most frequent and functional polyadenylation signal in MDV. The identification of possible alternative transcript termination (ATT) needs further experimental evidence (Figure 2 and Appendix A). ATT is a strong regulatory factor in eukaryotes [82], but there is only limited data for ATT in herpesvirus transcription.

The comparison of the transcriptome between the very virulent RB1B strain and the CVI988 vaccine revealed differences in only a few transcripts (Figure 1). However, more work needs to be done to unravel significant differences that could possibly point towards a mechanism of attenuation or provide valuable information for the development of diagnostic tools.

In summary, our data demonstrate that the MDV genome is more complex than previously assumed. It provides a source of reference for MDV transcripts expressed in primary chicken B cells and lays the foundation for future research on MDV-encoded gene products and splice variants.

## Figures and Tables

**Figure 1 viruses-11-00264-f001:**
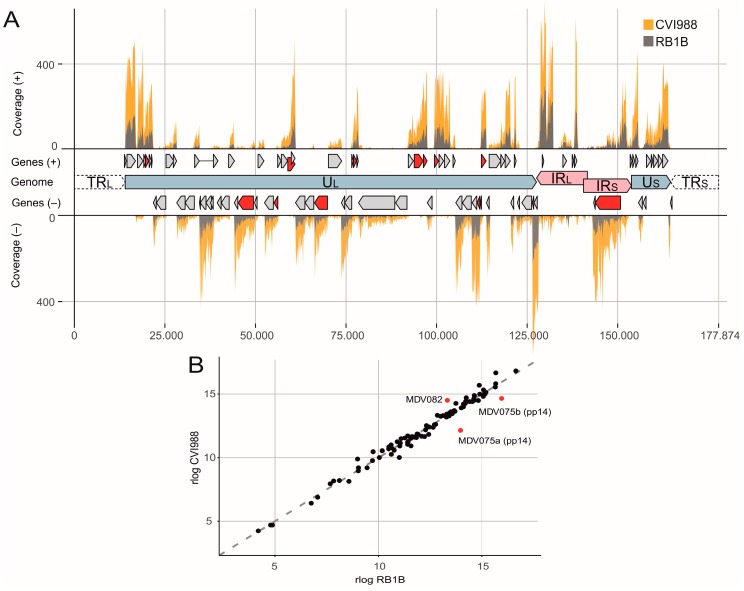
The Marek’s disease virus (MDV) transcriptome and proteome in in vitro infected primary chicken B cells. (**A**) Visualization of the deduplicated and strand-specific RNA-seq read coverage of plus (+) and minus (-) strand-encoded genes across the MDV genome. Orange curves indicate CVI988 reads and grey curves indicate reads for RB1B (with the respective annotated genes as grey arrow bars). Red bars depict proteins identified by MS. The two unique regions, unique long (U_L_) and short (U_S_) are flanked by terminal (TR_L_ and TR_S_) and internal (IR_L_ and IR_S_) inverted repeat regions. Nucleotide position numbers are derived from [12]. (**B**) Gene expression scatterplot comparing normalized expression levels (rlog) in RB1B and CVI988 infected primary chicken B cells. Red dots indicate significantly differentially expressed genes.

**Figure 2 viruses-11-00264-f002:**
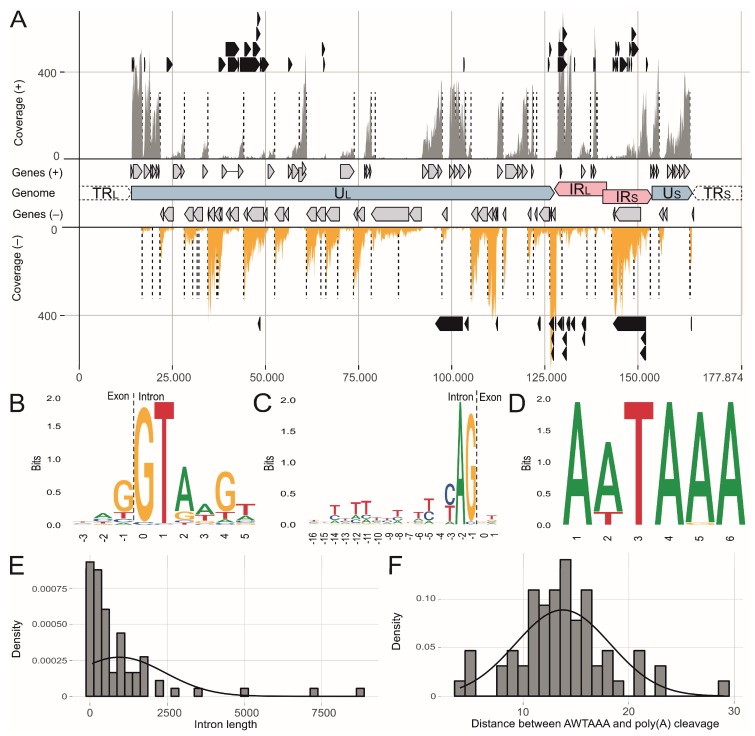
Overview of Marek’s disease virus (MDV) mRNA splicing and poly(A) cleavage. (**A**) Visualization of the cumulative CVI988 and RB1B RNA-seq coverage of plus (+) and minus (-) strand-encoded genes across the MDV genome. Black arrow bars indicate introns and dashed lines indicate poly(A) cleavage sites. More detailed information is shown in Appendix A and Appendix A. (**B**) Nucleotide frequency maps (sequence logo) of splice donor sites in MDV-encoded mRNAs. The relative heights of letters correspond to frequencies of bases at each position. (**C**) Sequence logo of splice acceptor sites in MDV-encoded mRNAs. (**D**) Sequence logo of polyadenylation signals in MDV-encoded three prime untranslated regions (3′-UTR). (**E**) Histogram depicting MDV intron length distributions. (**F**) Histogram depicting the distance from AWTAAA-like motifs to the poly(A) cleavage site.

**Figure 3 viruses-11-00264-f003:**
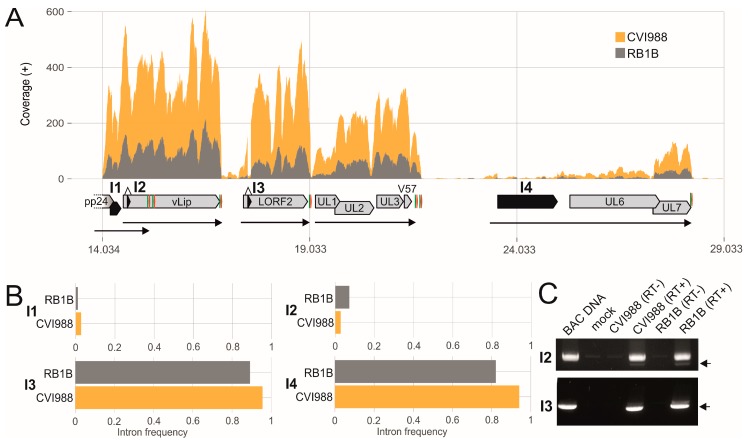
The Marek’s disease virus (MDV) unique long (U_L_) region. (**A**) Visualization of RNA-Seq coverage across parts of the MDV U_L_ region with respective introns in black. Green and red arrows indicate the polyadenylation signal and the poly(A) cleavage site respectively. Underlying black arrows suggest unspliced (mono-, bi-, or polycistronic) mRNAs. (**B**) Comparison of intron frequencies in RB1B and CVI988 infected primary chicken B cells. I1: pp24, I2: vLIP, I3: LORF2 (MDV012), I4: transcript antisense to UL5 (MDV017). (**C**) RT-PCR was performed to validate the splicing event. PCR products were derived using forward/reverse primers to amplify the respective intron-flanking regions. The representative gel images illustrate the results of RT-PCR analysis. The black arrows indicate the spliced form of the respective gene.

**Figure 4 viruses-11-00264-f004:**
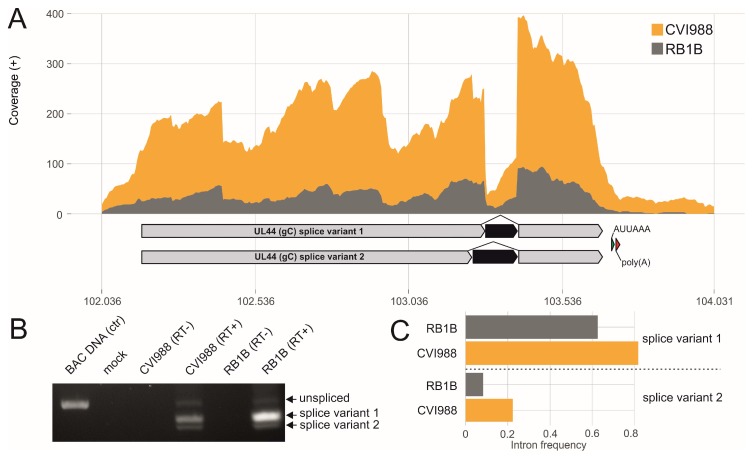
Glycoprotein C (gC) splicing. (**A**) Visualization of RNA-seq coverage across the Marek’s disease virus (MDV) gC gene UL44 with respective introns in black. The green and red arrow indicate the canonical ATTAAA polyadenylation signal and the poly(A) cleavage site respectively. (**B**) RT-PCR was performed to validate gC splicing. PCR products were derived using forward/reverse primers to amplify the respective intron-flanking regions. (**C**) Comparison of the gC intron frequencies in RB1B and CVI988 infected primary chicken B cells.

**Figure 5 viruses-11-00264-f005:**
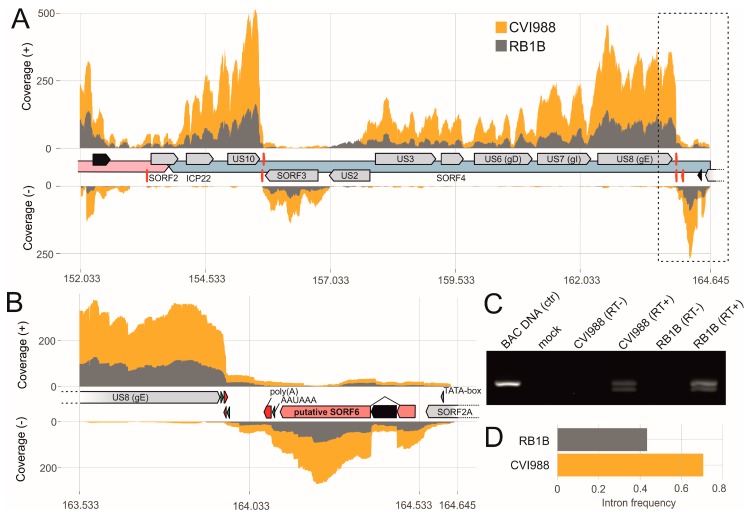
The Marek’s disease virus (MDV) unique short (U_S_) region. (**A**) Visualization of RNA-seq coverage across the MDV U_S_ region with respective introns in black. (**B**) Zoom into the far-right region of the MDV US with depiction of the novel U_S_ gene SORF6. Green and red arrows indicate the polyadenylation signals and the poly(A) cleavage sites, respectively. (**C**) RT-PCR was performed to validate the splicing event in the novel gene SORF6. PCR products were derived using forward/reverse primers to amplify the respective intron-flanking regions as full-length (upper band) and spliced (lower band). (**D**) Comparison of the novel U_S_ gene (SORF6) intron frequencies in RB1B and CVI988 infected primary chicken B cells.

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
