# Peer review of "The Transcriptional Landscape of Marek’s Disease Virus in Primary Chicken B Cells Reveals Novel Splice Variants and Genes"

_viruses, 2019, doi:10.3390/v11030264_

Reviewer 1 Report

This manuscript describes virus transcriptomes of virulent and attenuated (vaccine strain) MDV in their natural host cell type, chicken B cells. Transcriptional analyses during lytic infection of these viruses were mostly performed in primary fibroblasts, which are not infected naturally in vivo, previously. Thus, transcriptome revealed in the natural host cell type in this manuscript is highly informative. The experiments were performed carefully, and the data analyzed extensively. I have only a few very minor questions and suggestions as follows:

Line 31: The words “polycistronic MDV genome” sound strange to me. Almost all genomes including virus genomes are polycistronic anyway.

Lines 191-192 “Samples…”: Is this sentence complete? Conventional what? Formaldehyde?

Fig 5 legend: I cannot see green and red arrows in (A), but they are in (B). The explanation for these arrows should be moved to under (B), if this is the case. Also, is (B) a zoom in to the far left, not right, region of US in (A)?

Lines 413 to 422: The conclusion should be shorter. The first paragraph is mostly mere repetition of the introduction and unnecessary. 

Author Response

Dear Reviewer 1, we would like to thank you for the evaluation of our manuscript. We have addressed all of your concerns modified the manuscript as suggested, thereby improving its quality. You can find our direct responses to all comments below.

This manuscript describes virus transcriptomes of virulent and attenuated (vaccine strain) MDV in their natural host cell type, chicken B cells. Transcriptional analyses during lytic infection of these viruses were mostly performed in primary fibroblasts, which are not infected naturally in vivo, previously. Thus, transcriptome revealed in the natural host cell type in this manuscript is highly informative. The experiments were performed carefully, and the data analyzed extensively. I have only a few very minor questions and suggestions as follows:

Line 31: The words “polycistronic MDV genome” sound strange to me. Almost all genomes including virus genomes are polycistronic anyway.

·         We agree with the reviewer and removed “polycistronic” in line 31.

Lines 191-192 “Samples…”: Is this sentence complete? Conventional what? Formaldehyde?

·         Thanks for picking this up. We modified the indicated sentence to  “Samples were differentially labeled by dimethylation [44] using unlabeled and 13C-labeled formaldehyde, and…”

Fig 5 legend: I cannot see green and red arrows in (A), but they are in (B). The explanation for these arrows should be moved to under (B), if this is the case. Also, is (B) a zoom in to the far left, not right, region of US in (A)?

·         Thank you for the constructive comment. The Figure legend of Figure 5 has been corrected.

Lines 413 to 422: The conclusion should be shorter. The first paragraph is mostly mere repetition of the introduction and unnecessary.

·         Thank you for the suggestion – we’ve slightly shortened the Conclusion part of the manuscript (line 415ff).

We hope that, with the additional modifications made, the manuscript will be favorably reviewed and be acceptable for publication in Viruses.

Reviewer 2 Report

In this intriguing study, the authors have performed RNA-seq on a virulent MDV strain (RB1B) and compared it to a vaccine strain (CVI988/Rispens).  RNA-seq was performed in primary chicken B cells, which are biologically relevant.  Interestingly, the authors provide strong evidence that the MDV genome is much more complicated than thought, in part because extensive splicing occurs.  For example, gC was spliced, leading to a gC protein that does not contain a transmembrane domain and is secreted.  Other studies indicated that the ICP0 homologue in MDV was not expressed, which is very surprising.  The RNA-seq studies were also confirmed by various means, which is crucial for validating RNA-seq studies.  Although this is an important manuscript that is well written and describes interesting findings, I have a couple of suggestions that are summarized below.

1.      Lines 256-261 basically states that only minor differences were observed between in RNA-seq studies were observed in the virulent versus vaccine strain.  This is surprising.  I was wondering do the authors feel these minor differences are important? Or do the authors feel the important differences are in protein changes in specific virulence determinants or is there another cell-type that might allow bigger differences to be observed in RNA-seq studies.  Hence, a bit more discussion about what this surprising result means would be valuable.

2.     Additional discussion about the polyA+ spliced LATs would be useful and whether the ORFs in these LATs appear as if they may be expressed.  Since it is assumed that LAT is a non-coding transcript, this information is important.

3.     Is there any evidence that ICP0 is expressed in any cell by MDV, as this result was surprising?  Does the ICP0 ORF look like a true ORF and contain a nuclear localization signal or a C3HC4 zinc RING finger located near the amino terminus of other ICP0 proteins?  This information would strengthen their exciting finding. 

Author Response

Dear reviewer 2, we would like to thank you for the valuable evaluation of our manuscript. We have addressed all of your concerns modified the manuscript as suggested, thereby improving its quality. You can find our direct responses to all comments below.

In this intriguing study, the authors have performed RNA-seq on a virulent MDV strain (RB1B) and compared it to a vaccine strain (CVI988/Rispens).  RNA-seq was performed in primary chicken B cells, which are biologically relevant.  Interestingly, the authors provide strong evidence that the MDV genome is much more complicated than thought, in part because extensive splicing occurs.  For example, gC was spliced, leading to a gC protein that does not contain a transmembrane domain and is secreted.  Other studies indicated that the ICP0 homologue in MDV was not expressed, which is very surprising.  The RNA-seq studies were also confirmed by various means, which is crucial for validating RNA-seq studies.  Although this is an important manuscript that is well written and describes interesting findings, I have a couple of suggestions that are summarized below.

1.      Lines 256-261 basically states that only minor differences were observed between in RNA-seq studies were observed in the virulent versus vaccine strain.  This is surprising.  I was wondering do the authors feel these minor differences are important? Or do the authors feel the important differences are in protein changes in specific virulence determinants or is there another cell-type that might allow bigger differences to be observed in RNA-seq studies.  Hence, a bit more discussion about what this surprising result means would be valuable.

•          Thank you for the constructive comment. We extended the discussion to highlight this surprising finding (line 250ff).

2.     Additional discussion about the polyA+ spliced LATs would be useful and whether the ORFs in these LATs appear as if they may be expressed.  Since it is assumed that LAT is a non-coding transcript, this information is important.

·                The reviewer raises an important point. We sequence polyA-enriched and thereby identified polyA+ RNA that correspond to precursor miRNAs previously identified in the LAT locus. Full length, polyA- RNA was not detected due to the RNA enrichment method.

3.     Is there any evidence that ICP0 is expressed in any cell by MDV, as this result was surprising?  Does the ICP0 ORF look like a true ORF and contain a nuclear localization signal or a C3HC4 zinc RING finger located near the amino terminus of other ICP0 proteins?  This information would strengthen their exciting finding.

·           Thanks for raising this interesting point. In addition to our experimental data, we performed in silico predictions and observed that the MDV ICP0 homologue does not contain a nuclear localization signal or a C3HC4 zinc RING finger located near the amino terminus. We included this information in the manuscript (line 388f). To our knowledge, there is no evidence for MDV ICP0 expression in the literature.  However, the RLORF1 (ICP0-like protein) is annotated as such and has been identified as ORF in the TRL and IRL.  It appears that the absence of ICP0 is a feature of avian alphaherpesviruses.

We hope that, with the additional modifications made, the manuscript will be favorably reviewed and be acceptable for publication in Viruses.
